Genome-wide identification of HSF family in peach and functional analysis of PpHSF5 involvement in root and aerial organ development

Tan Bin
Yan Liu
Li Huannan
Lian Xiaodong
Cheng Jun
Wang Wei
Zheng Xianbo
Wang Xiaobei
Li Jidong
Ye Xia
Zhang Langlang
Li Zhiqian
Feng Jiancan jcfeng@henau.edu.cn
1 College of Horticulture, Henan Agricultural University , Zhengzhou , China
2 Henan Key Laboratory of Fruit and Cucurbit Biology , Zhengzhou , China
Maloof Julin
Electronic publication date: 2021 Mar 12
Publication date: 2021
Volume: 9
Electronic Location ID: e10961
Received 2020 Feb 13; Accepted 2021 Jan 27
Copyright: ©2021 Tan et al.
Copyright year: 2021
Copyright holder: Tan et al.
License: This is an open access article distributed under the terms of the Creative Commons Attribution License, which permits unrestricted use, distribution, reproduction and adaptation in any medium and for any purpose provided that it is properly attributed. For attribution, the original author(s), title, publication source (PeerJ) and either DOI or URL of the article must be cited.
License URL: https://creativecommons.org/licenses/by/4.0/

Keywords: Heat shock factors family, Root development, Peach (Prunus persica), Functional identification, PpHSF5, Aerial organ

Funding: National Key Research and Development Program of China 2018YFD1000101 Modern Agricultural Industry Technology of Henan province S2014-11-G02 Innovation Team project of Henan University 19IRTSTHN009 This work was supported by the National Key Research and Development Program of China (2018YFD1000101), by the Modern agricultural industry technology of Henan province (S2014-11-G02) and by the Innovation Team project of Henan University (19IRTSTHN009). The funders had no role in study design, data collection and analysis, decision to publish, or preparation of the manuscript.

==============================
Background

Heat shock factors (HSFs) play important roles during normal plant growth and development and when plants respond to diverse stressors. Although most studies have focused on the involvement of HSFs in the response to abiotic stresses, especially in model plants, there is little research on their participation in plant growth and development or on the HSF (PpHSF) gene family in peach (Prunus persica).

Methods

DBD (PF00447), the HSF characteristic domain, was used to search the peach genome and identify PpHSFs. Phylogenetic, multiple alignment and motif analyses were conducted using MEGA 6.0, ClustalW and MEME, respectively. The function of PpHSF5 was confirmed by overexpression of PpHSF5 into Arabidopsis.

Results

Eighteen PpHSF genes were identified within the peach genome. The PpHSF genes were nonuniformly distributed on the peach chromosomes. Seventeen of the PpHSFs (94.4%) contained one or two introns, except PpHSF18, which contained three introns. The in silico-translated PpHSFs were classified into three classes (PpHSFA, PpHSFB and PpHSFC) based on multiple alignment, motif analysis and phylogenetic comparison with HSFs from Arabidopsis thaliana and Oryza sativa. Dispersed gene duplication (DSD at 67%) mainly contributed to HSF gene family expansion in peach. Promoter analysis showed that the most common cis-elements were the MYB (abiotic stress response), ABRE (ABA-responsive) and MYC (dehydration-responsive) elements. Transcript profiling of 18 PpHSFs showed that the expression trend of PpHSF5 was consistent with shoot length changes in the cultivar ‘Zhongyoutao 14’. Further analysis of the PpHSF5 was conducted in 5-year-old peach trees, Nicotiana benthamiana and Arabidopsis thaliana, respectively. Tissue-specific expression analysis showed that PpHSF5 was expressed predominantly in young vegetative organs (leaf and apex). Subcellular localization revealed that PpHSF5 was located in the nucleus in N. benthamiana cells. Two transgenic Arabidopsis lines were obtained that overexpressed PpHSF5. The root length and the number of lateral roots in the transgenic seedlings were significantly less than in WT seedlings and after cultivation for three weeks. The transgenic rosettes were smaller than those of the WT at 2–3 weeks. The two transgenic lines exhibited a dwarf phenotype three weeks after transplanting, although there was no significant difference in the number of internodes. Moreover, the PpHSF5-OE lines exhibited enhanced thermotolerance. These results indicated that PpHSF5 might be act as a suppresser of growth and development of root and aerial organs.

Introduction

Plant growth and development are affected by a range of abiotic stress, including cold, heat, salinity and drought stress (Guo et al., 2016). Heat shock factors (HSFs) act with heat shock proteins (HSPs) as key transcriptional activators during responses to abiotic stress (Hu, Hu & Han, 2009). Recent studies indicated that HSFs act as key components of signal transduction in response to different abiotic stresses in plants (Guo et al., 2016; Scharf et al., 2012).

HSFs in plant genomes can be identified by a conserved DNA-binding domain (DBD). The DBD domain is located in the N-terminal of all HSFs and specifically binds to heat stress (HS) motifs in the promoters of target genes (Wang et al., 2018). The adjacent HR-A/B region is linked to the DBD by a connector of variable length (15–80 amino acid residues) that contains a bipartite heptad pattern of hydrophobic amino acid residues, which constitutes a coiled-coil domain for protein interaction. According to the number of amino acid residues inserted into the HR-A/B region, HSFs are divided into three main groups, each with subgroups, namely HSFA (A1-A9), HSFB (B1-B5) and HSFC (C1-C2) (Koskull-Doring, Scharf & Nover, 2007; Yang et al., 2014). HSFA members contain an acidic motif (AHA activation domain) at their C-terminus and act as transcriptional activators. The members of HSFB act as transcriptional repressors.

In a wide range of plants, a number of HSFs have been shown to be involved in resistance to heat (Guo et al., 2016) and other abiotic or biotic stresses (Yu et al., 2019). Of the 21 HSF family members in Arabidopsis, a number act as pioneer regulators of the response to heat shock. HSFA1a, HSFA1b, HSFA1d, HSFA1e and HSFA2 play active regulatory roles in the response to HS in plants (Busch, Wunderlich & Schoffl, 2005; Nishizawa et al., 2006). In Arabidopsis, the assembly of the HSFA1/A2 super-activated complex regulates heat stress genes (Chan-Schaminet et al., 2009). HSFB1 and HSFB2b participate in disease resistance regulation of Arabidopsis and expression of Pdf1.2 (Kumar et al., 2009). OsHSFB4b and OsHSFA2c participate in the regulation of the heat shock response by regulating the expression of HSP100 (Singh et al., 2012). OsHSFC1b is related to the regulation of salt stress and plant development (Schmidt et al., 2012).

Several HSFs are stress-inducible transcriptional factors that participate in the growth and development of root and aerial organs in plant. Overexpression of AtHsfB4 in Arabidopsis induces specific effects on root development, resulting in shortened roots (Begum, Reuter & Schoffl, 2013). The over-expression of BhHsf1 conferred growth retardation of aerial organs, producing a dwarf phenotype, although the primary roots were not obviously different from those of wild type (Zhu et al., 2009). Transgenic Arabidopsis plants with strong expression of AtHsfA3 and AtHsfA2 showed a severely dwarfed phenotype and increased tolerance to heat (Ogawa, Yamaguchi & Nishiuchi, 2007; Yoshida et al., 2008). The thermotolerant phenotype was also observed in the cotyledons, rosette leaves, inflorescence stems and seeds of transgenic Arabidopsis plants expressing OsHsfA2e (Yokotani et al., 2007).

The HSF family have been analyzed genome-wide in several plants, such as rice (Oryza sativa), Arabidopsis (Arabidopsis thaliana), cotton (Gossypium hirsutum), soybean (Glycine max), wheat (Triticum aestivum), pepper (Capsicum annuum), poplar (Populus trichocarpa), Brassica napus, grape (Vitis vinifera) and Tartary buckwheat (Fagopyrum tataricum) (Nover et al., 2001; Chauhan et al., 2011; Wang et al., 2014; Li et al., 2014; Guo et al., 2015; Xue et al., 2014; Zhang et al., 2016; Zhu et al., 2017; Liu et al., 2018; Liu et al., 2019). Peach (Prunus persica L.) is an important economical crop and a popular fruit with consumers. However, there are limited studies on peach HSFs. To remedy this, we identified the HSF family in the peach genome and conducted bioinformatics analysis of the 18 identified PpHSFs. Based on the latest transcriptome data (Lian et al., 2020), the expression patterns of the PpHSF genes were analyzed during development of the cultivar ‘Zhongyoutao 14’. ‘Zhongyoutao 14’ (derived from ‘SD9238’), is a temperature-sensitive peach that exhibits a shorter internode length and a smaller canopy when grown below 30 °C (Lu et al., 2016). PpHSF5 was further analyzed and found to function in the development of the root and aerial organs. Furthermore, the thermotolerant phenotype was analyzed in newly obtained transgenic Arabidopsis plants expressing PpHSF5. The genome-wide analysis of PpHSF gene family offers a basis for further investigation into the function and evolutionary history of peach HSFs and provides candidate genes for peach molecular breeding.

Materials & Methods

Plant materials

Established peach trees (Prunus persica) cultivar ‘Zhongyoutao 14’ (‘Maotao’ as rootstock) have been grown for 5 years at the Experimental Station of the Horticulture College, Henan Agricultural University (Zhengzhou, China). Samples from the apex, young and mature leaves, self-pollinated embryos, and fruit were collected, frozen in liquid nitrogen and stored at −80 °C. Leaves from Nicotiana benthamiana were used for subcellular location of PpHSF5. Arabidopsis thaliana (L.) Heynh Columbia 0 (Col-0) was used for transformation with PpHSF5.

Identification and chromosomal location of HSF genes in peach

The hidden Markov model (HMM) of the DBD domain (PF00447), characteristic of HSFs, was downloaded from the Pfam website (Finn et al., 2000) and used to identify HSF genes in peach. The peach genome files (v2.1) were downloaded from JGI database (https://phytozome.jgi.doe.gov/pz/portal.html) (Verde et al., 2013), HSF protein sequences were obtained in peach genome by BLASTP and hmmsearch function, and then the DBD domain were further identified by Pfam analysis. The peach HSF gene and protein sequences were extracted from Phytozome v12.1. PpHSF genes were named according to physical location on the chromosomes. Positional information was retrieved from peach genome annotations obtained from Phytozome v12.1, and chromosome locations of the PpHSFs were drawn using the Circos software (Krzywinski et al., 2009). The isoelectric points and other physical properties were approximated from ExPASy (http://web.expasy.org/compute_pi). Gene structures were predicted using the Gene Structure Display Server 2.0 (http://gsds.cbi.pku.edu.cn/).

Phylogenetic and motif analysis of PpHSFs

The amino acid sequences of 21 AtHSFs (Arabidopsis thaliana), 25 OsHSFs (Oryza sativa) and 18 PpHSFs (Prunus persica) were gathered from Phytozome v12.1 using ClustalW with system default settings. The phylogenetic trees were formulated by the maximum likelihood method (ML) with Jones-Taylor-Thornton (JTT) model in MEGA 6.0 (http://www.megasoftware.net/download_form).. Conserved motifs of HSF proteins in peach were identified using the MEME tool (http://meme.nbcr.net/meme/cgi-bin/meme.cgi) with default parameters in normal operation mode. The subcellular localization was predicted with Plant-mPLoc (http://www.csbio.sjtu.edu.cn/bioinf/plant-multi/#).

Gene duplication and Cis -element analysis of PpHSFs

Gene duplication was analyzed using MCScanX (Wang et al., 2012). Genomic DNA sequences (2000 bps upstream of the start codons) for each PpHSF were obtained from the peach genome and skimmed in the PlantCARE database (http://bioinformatics.psb.ugent.be/webtools/plantcare/html/) for cis-acting elements analysis of the promoter in PpHSFs.

Gene expression analysis of PpHSFs

The FPKM (fragments per kilobase of exon per million fragments mapped) values of the 18 PpHSFs (Table S1-1) were obtained from our previous study of gene expression in shoots at four critical growth stages, namely initial period (IP), initial elongation period (IEP), rapid growth period (RGP) and stable growth period (SGP) of temperature-sensitive peach cultivar ‘Zhongyoutao 14’ (Lian et al., 2020). The average maximum temperature of previous week (AMTPW) began to be higher than 30 °C in the first day of RGP (Lian et al., 2020). The heat map was generated by TBtools (Chen et al., 2020).

Quantitative real-time PCR analysis of PpHSF5

Total RNA of different tissues from ‘Zhongyoutao 14’ peach and leaves from T2 transgenic Arabidopsis lines was isolated using the Spin Column Plant Total RNA Purification Kit (ShengGong, Shanghai, China). The cDNA was synthesized using FastQuant RT Kit (Tiangen Biotech, Beijing, China). qRT-PCR was implemented using an ABI PRISM 7500 FAST Sequence Detection System (Applied Biosystems, Madrid, CA, USA) with SYBR Select Master Mix (Applied Biosystems, USA). Primers of PpHSF5 were designed using Primer Premier 5.0. PpGAPDH (Prupe.1G234000) and AtUBC (AT5G25760) were used as constitutive controls for either tissue-specific expression in peach or expression analysis in transgenic Arabidopsis, respectively. Primers are shown in Table S1-2. The reaction mixture was as follows: 1 µL cDNA template (200 ng/ µL), 1 µL of each primer (10 µM), 10 µL SYBR Premix and 7 µL ddH2O. Melting curve analysis was performed after the end of 40 cycles to insure proper amplification of the target. During the melting process, fluorescence readings were continuously collected from 60−90 °C at a heating rate of 0.5 °C s−1. All analysis was repeated three times using biologically replicates. The relative expression levels of PpHSF5 were calculated as 2−ΔΔCT method (Schmittgen & Livak, 2008). The relative expression levels of PpHSF5 was calculated in SPSS using ANOVA at significance levels of P < 0.05.

Subcellular localization of PpHSF5

PpHSF5 without the termination codon was amplified by PCR using cDNA from ‘Zhongyoutao 14’ as the template (Primer details in Tables S1–S2). This coding region was cloned into the pSAK277-GFP vector to construct PpHSF5::GFP fusion proteins that were driven by the CaMV 35S promoter. The p35SPpHSF5::GFP and p35SGFP (control) vectors were transformed into Agrobacterium tumefaciens strain GV3101, which were then injected into leaves of N. benthamiana according to previously published protocols (Sparkes et al., 2006). The leaves were observed 48–72 h after injection using laser scanning confocal microscopy (Zeiss LSM700).

Construction of expression vectors for plant transformation

The CDS of PpHSF5 was PCR-amplified and cloned into the pSAK277 vector using the restriction enzymes Xho I and Xba I (Primer details in Table S1–S2). The p35S::PpHSF5 vector was transformed into Agrobacterium tumefaciens strain GV3101. The floral dip method was used to transform Arabidopsis thaliana (Col-0) (Chung, Chen & Pan, 2000).

Phenotype of overexpression PpHSF5 in Arabidopsis

The seeds from T2 transgenic Arabidopsis lines were sterilized by 6.25% NaClO for 5 min, and then washed in sterilized ddH2O. The seeds were cultured at 4 °C for 2 d and then transferred onto MS solid medium under 16/8 h light/dark cycle for one week on square plates. Three biological replicates (with three seedlings of each lines per square plate) were used for observation of root phenotype. The roots of different transgenic lines with three plants per line were measured by a LA2400 Scanner at three weeks to determine the growth status. The leaves were cut from the transgenic seedlings for gene expression analysis. Other seedlings, germinated on agar and grown for one week, were transferred into the soil and grown under normal conditions. The length and width of rosettes (four leaves per plants, five repetition) and number of rosettes (five plants per line) in different transgenic lines and WT were measured and photographed at two weeks and at three weeks after transplanting, respectively. Moreover, the morphology of transgenic lines and WT, including the height of plants (eight plants per line) and the number of branches and blooms (five plants per line) were recorded, three weeks after transplanting.

Heat stress treatment

For performing heat stress treatment on the seeds germination and plants grown on the agar medium, seeds of WT and transgenic Arabidopsis lines sown on MS medium at 4 °C for 2 d and in darkness for 2 d (22 °C) were exposed to HS stress at 46 °C for 30 min, and then were transferred into a climate chamber (22 °C, 16 h light/8 h dark cycles). After HS treatment, the germination of seeds were counted daily and photographed. More than 50 seeds of each line were used in each plate with three replications. Difference in HS stress was confirmed using t-test.

Statistical analysis

Data were analyzed by ANOVA, Tukey HSDa and Duncana’s multiple range tests (at P < 0.05) using IBM SPSS Statistics 20 (SPSS, USA).

Results

Genome-wide identification, chromosomal distribution and gene structures of HSF genes in peach

Eighteen HSF gene family members were identified from the peach genome and then named PpHSF1 to PpHSF18 according to their physical locations (Table 1 and Table S2-1). The PpHSF coding sequences ranged from 591 bp (PpHSF8) to 1608 bp (PpHSF14). In silico-translated PpHSF proteins showed divergent lengths [196 to 535 amino acids (aa)] with different molecular weights (22.36∼59.56 kDa) and isoelectric points (4.67 to 8.75) (Table1). All PpHSFs were predicted to be nuclear-localized proteins.

Table 1 Basic information of PpHSF gene family members.

Gene name	Gene ID	Length of CDS (bp)	No. of amino acids (aa)	Molecular weight (Da)	Predicted isoelectric point (PI)	Chromosome location	Subcellular localization	
PpHSF1	Prupe.1G021200	1068	355	41166.82	4.9	1	Nucleus	
PpHSF2	Prupe.1G165500	1452	483	54128.99	5.61	1	Nucleus	
PpHSF3	Prupe.1G335100	1227	408	46017.53	4.81	1	Nucleus	
PpHSF4	Prupe.1G410400	1125	374	41952.84	4.95	1	Nucleus	
PpHSF5	Prupe.1G433500	1170	389	43557.76	7.17	1	Nucleus	
PpHSF6	Prupe.2G292100	912	303	33900.98	5.19	2	Nucleus	
PpHSF7	Prupe.3G108700	1314	437	49855.78	5.13	3	Nucleus	
PpHSF8	Prupe.4G046000	591	196	22364.46	8.75	4	Nucleus	
PpHSF9	Prupe.4G068100	1224	407	46118.31	5.23	4	Nucleus	
PpHSF10	Prupe.4G144200	1512	503	56052.61	4.78	4	Nucleus	
PpHSF11	Prupe.5G031100	1551	516	56261.49	4.67	5	Nucleus	
PpHSF12	Prupe.5G093200	996	331	36068.73	4.75	5	Nucleus	
PpHSF13	Prupe.7G056700	735	244	28001.34	5.75	7	Nucleus	
PpHSF14	Prupe.7G117200	1608	535	59567.89	4.98	7	Nucleus	
PpHSF15	Prupe.7G133600	900	299	33339.83	5.07	7	Nucleus	
PpHSF16	Prupe.7G206900	1458	485	53632.01	5.07	7	Nucleus	
PpHSF17	Prupe.7G231100	1002	333	37851.52	5.68	7	Nucleus	
PpHSF18	Prupe.8G234900	1080	359	40936.56	5.58	8	Nucleus	

Seven of the 8 peach chromosomes contained at least one PpHSF, with the exception being chromosome 6 (Fig. 1 and Table 1). Five PpHSFs were located on chromosome 1 (PpHSF1-5), and another five (PpHSF13-17) on chromosome 7. Chromosomes 2, 3 and 8 carried only one PpHSF gene each, while chromosome 5 had two, and chromosome 4 had three. The above results indicated that PpHSFs were unevenly distributed across the peach chromosomes.

Figure 1 Chromosomal location of HSF genes in peach (PpHSFs).

Three syntenic pairs are linked by red lines.

The structural differences of the PpHSF genes were also analyzed. The number of introns ranged from one to three among the PpHSFs. The majority of the PpHSFs (66.67%) contained one intron, 27.78% contained two introns, and only PpHSF18 contained three introns (Fig. S1 and Table S2). Interestingly, both PpHSF18 and PpHSF12 has predicted introns in the 5′-UTR and 3′-UTR, respectively.

Gene duplication pattern analysis of PpHSFs

To explain the expansion of the PpHSFs gene family, the gene duplication patterns of the PpHSFs were analyzed and compared across the peach genome (Table S3). There were only two patterns of gene duplications, with 67% of the gene pairs derived from dispersed gene duplication (DSD) and the remaining gene pairs derived from whole-genome duplication (WGD). Three syntenic pairs were identified, and all originated from WGD. The syntenic genes were located on different chromosomes from their partner (Fig. 1).

Classification, phylogenetic and motif analyses of PpHSFs

Among plant species, there are two characteristic amino acid domains in the HSF family, the DBD and adjacent HR-A/B region (Nover et al., 2001). The PpHSFs were divided into three classes (PpHSFA, PpHSFB and PpHSFC), according to the number of amino acids between part A and part B of the HR-A/B domain (Fig. 2A). Multiple sequence alignment analysis of the PpHSF proteins indicated that an insertion of 21 amino acids was found in Class A (11 of the PpHSFs) and a shorter insertion of 14 amino acid in Class C (1 of the PpHSFs) between the HR-A and HR-B regions. Six of the PpHSFs had no aa insertion between the two domains (Class B).

Figure 2 Multiple sequence alignment of the HR-A/B regions (OD), conserved motif and phylogenetic analysis of PpHSFs.

(A) Multiple sequence alignment of the HR-A/B regions, from the start of the DNA-binding domain to the end of the HR-A/B region, of the HSF proteins were aligned with MEGA 6. (B) Hylogenetic tree of HSFs from Prunus persica (Pp, red star), Oryza sativa (Os, blue circle) and Arabidopsis thaliana (At, green square) constructed by maximum likelihood method (ML) with Jones-Taylor-Thornton (JTT) model in MEGA 6.0. Both locus ID and subclass numbers are listed. (C) Analysis of conserved motifs in the HSF gene family in peach. Proteins are organized according to the groups in Fig. 2A. Ten motifs were found in the protein sequences as shown in Table S4.

Phylogenetic analysis among the HSF proteins from three plant species, namely 21 AtHSFs (Arabidopsis thaliana), 25 OsHSFs (Oryza sativa) and 18 PpHSFs (Prunus persica), was conducted by constructing a phylogenetic tree. According to the phylogenetic tree, the 64 HSFs derived from the three plant species were divided into three classes and 15 subclasses (Fig. 2B). The peach proteins sorted into the classes of HSFs, within (11 members) in class HSFA, six in HSFB, and one in HSFC. Class A included nine subclasses (A1-A9), the largest number of subclasses. The PpHSFs were grouped into eight of the Class A subclasses, with no PpHSF in Class A7. Class B consisted of 18 total members and was divided into four subclasses (B1-B4). It is noteworthy that PpHSF8 clustered with Class B but as a single branch. Only six members were clustered into Class C, with two subclasses (C1-C2). No PpHSFs clustered with subclass C2.

The conserved motifs in the PpHSF proteins were analyzed using MEME. The results revealed that PpHSFs contained ten conserved motifs (Fig. 2C and Table S4). Motifs 1-3 were found in the N-terminals (the most conserved region) of each PpHSF. Motif 4 was found in Class A and Class B. Motif 5, which was found between the HR-A and HR-B regions, was observed in Class A and Class C. The motif analysis was consistent with the multiple sequence alignment and phylogenetic analyses.

Analysis of the Cis-acting regulatory elements in the PpHSF gene promoters

The cis-acting elements within the promoters of the 18 PpHSFs were analyzed using PlantCARE. Every promoter contained at least two MYB elements (abiotic stress response) (Table 2). All but one promoter contained an ABRE (ABA-responsive element). The next most common elements were MYC elements (dehydration-responsive) (in 88.8% of the promoters), CGTCA- and TGACG- motifs (83.3%), and ARE elements (anaerobic induction) (77.8%). ERE (ethylene-responsive element), MBS (drought inducible), MRE and P-box elements were also present in the promoters of some PpHSFs. The TCA-motif was observed in only five PpHSFs, namely PpHSF1, PpHSF2, PpHSF5, PpHSF6 and PpHSF13. Previous studies reported several elements, including MYB, ABRE, MYC, play vital roles in stress responses in plants (He et al., 2012; Li et al., 2012). The different cis-elements in the promoter regions of these PpHSFs implied that the PpHSFs may function in plant development and stress responses.

The expression patterns of PpHSFs during shoot elongation in ‘Zhongyoutao 14’

Based on our previous RNA-seq analysis (Lian et al., 2020), the expression patterns of PpHSFs were compared in four critical stages of shoot elongation of ‘Zhongyoutao 14’ grown under elevated temperature in the field (Fig. 3). Most of PpHSFs belonging to the A and C classes (except PpHSF4 and PpHSF11) were maintained at lower expression level. The PpHSFs of B class exhibit diverse expression patterns. The FPKM values of PpHSF8 and PpHSF13 remained almost unchanged at the four stages. The transcripts of another three PpHSFs (PpHSF15, PpHSF6 and PpHSF12) were present at lower levels during the IEP stage and then slightly increased during the RGP and SGP stages. The expression of level of PpHSF5 showed higher in IEP stage and increased from the RGP to SGP stages. PpHSF5 might participate in temperature-induced shoot growth of temperature-sensitive peach.

Expression analysis of PpHSF5 and subcellular localization of PpHSF5

The relative expression of PpHSF5 was investigated by qRT-PCR in different organs of ‘Zhongyoutao 14’ (Fig. 4; Table S5-1). The results showed that PpHSF5 were expressed predominantly in young vegetative organs (leaves and apex), but barely detectable in embryos and mature leaves. This suggested that PpHSF5 might participated in the growth and development of plants. The 35S::PpHSF5-GFP signal was evident in the cellular nucleus in N. benthamiana cells, indicating a nuclear localization (Fig. 5). The result was in concurrence with the prediction from Plant-mPLoc of subcellular localization (Table 1).

Table 2 Cis-elements in the promoters of eighteen PpHSF genes.

	ABRE	ARE	CGTCA-motif	ERE	MBS	MRE	MYB	MYC	P-box	TGACG-motif	TCA-element	LTR	TGA-element	
PpHSF1	1	3	2	2	1	1	4	3	2	2	2	–	–	
PpHSF2	1	2	2	1	–	–	3	1	–	2	1	3	1	
PpHSF3	1	1	–	2	1	1	4	2	–	1	–	1	1	
PpHSF4	3	6	3	–	–	–	3	–	1	3	–	2	–	
PpHSF5	5	–	3	–	2	–	7	–	1	3	3	–	–	
PpHSF6	4	–	3	–	1	–	7	5	2	3	1	–	–	
PpHSF7	1	5	–	1	–	–	6	3	–	–	–	1‘	1	
PpHSF8	3	3	3	1	1	1	3	8	1	3	–	–	3	
PpHSF9	5	5	4	–	–	1	13	4	–	4	–	2	2	
PpHSF10	3	1	5	2	–	–	2	6	–	5	–	–	2	
PpHSF11	–	1	1	–	2	–	7	8	–	1	–	–	–	
PpHSF12	6	–	1	1	1	1	13	3	–	1	–	–	1	
PpHSF13	3	4	1	–	–	–	4	5	–	1	1	1	–	
PpHSF14	3	4	3	–	3	–	11	3	–	3	–	1	–	
PpHSF15	8	2	4	1	1	1	6	5	–	4	–	–	–	
PpHSF16	4	4	1	1	–	2	4	11	–	–	–	–	–	
PpHSF17	11	2	2	1	1	–	4	6	–	2	–	–	–	
PpHSF18	13	3	–	1	1	2	2	4	–	–	–	1	–	

Overexpression of PpHSF5 in arabidopsis results in dwarf phenotypes

To investigate the function of PpHSF5, an overexpression vector with PpHSF5 was transformed into Arabidopsis. The phenotype of two transgenic lines and WT were recorded (Fig. 6). One week after germination on agar medium, the transgenic lines had shorter roots and a smaller number of lateral roots than WT seedlings (Fig. 6A). The average root length in WT was 7.13 cm, in transgenic line L1 was 3.08 cm, and in L2 was 3.50 cm (Figs. 6A and 6B; Table S6-1). Two weeks after transplantation, there was no difference in the number of rosette leaves between the transgenic lines and WT (Fig. 6C and 6D–6A and Table S6-2), although the rosette leaves were significantly shorter and narrower in the transgenic lines (the average length and width; Figs. 6D–6B, 6C and Tables S6-3 and S6-4). The mRNA levels in the PpHSF5-OE lines were obviously higher than WT plants (Fig. 6D and Table S5-2).

Figure 3 Heatmap of transcript levels of HSF genes in peach.

Transcriptome data were used to measure the expression level of PpHSFs. The gene names on the right are organized according to the different subclasses. Samples were harvest from shoots at the IP (initial period), IEP (initial elongation period), RGP (rapid growth period), and SGP (stable growth period), which are four key growth stages during temperature-sensitive peach shoot development. Color scale at the top represents FPKM values. Blue indicates low expression and red indicates high expression. Heatmap was generated using TBtools.

Figure 4 Relative expression of PpHSF5 in different tissues of ‘Zhongyoutao 14’ peach.

Established plants were grown under normal conditions. The analyzed tissues include the apex, flower, embryo, young leaf, and mature leaf,which harvested at the same time. The relative expression levels were calculated using the 2−ΔΔCT method.

Figure 5 Subcellular localization of PpHSF5 in N. benthamiana epidermal cells.

(A and D) Images of green fluorescence from the GFP protein and the PpHSF5-GFP fusion protein in tobacco cells under the confocal microscope; (B and E) Bright field image of tobacco epidermal cells; (C) Overlay of A and B; (F) Overlay of D and E.

Three weeks after transplanting, the soil-grown transgenic lines had fewer rosette leaves and the leaves were shorter and narrower than those in WT plants (Fig. 6F). Moreover, the two transgenic lines (L1 and L2) exhibited a dwarf phenotype (Figs. 6E and 6F). The average height of L1 (16.83 cm) was 40% shorter than that of the WT (26.77 cm). The number of rosette branches was much greater in WT than in transgenic lines, which had just one flowering stalk (Fig. 6F). There was no significant difference in the number of internodes (Figs. 6E and 6F), indicating that the dwarf phenotype of the transgenic lines might be caused by shorter internode length.

Shorter roots were also observed in the transgenic lines for cultivation three weeks after transplanting (Figs. 6G and 6H). Root length and root volume were significantly lower in transgenic lines compared to WT (Figs. 6G and 6H, Tables S6-12, S6-13. The average length of roots in Line 1 was 219.34 cm, which was 54% of the length in WT plants. The root volumes in the transgenic lines (Line 1 was 0.19 cm3, Line 2 was 0.36 cm3) was no more than 20% of that in WT (1.95 cm3). Other root indexes output by the root scanner were also less in the PpHSF5-OE lines, including the forks, tips and crossings of roots (Figs. 6G and 6H, Tables S6-14, S6-15, S6-16). Between the two transgenic lines, the higher expression level of PpHSF5 in L1 resulted in more obvious phenotypes compared to PpHSF5-OE L2 and WT (Figs. 6A, 6C, and 6E and Table S5-2). The above results indicated that PpHSF5 might participate in plant growth and development and that overexpression of PpHSF5 results in a dwarf phenotype in transgenic Arabidopsis.

Figure 6 Phenotypic and expression analysis of transgenic Arabidopsis over-expressing PpHSF5.

(A) Phenotype of T2 transgenic plants from two lines over-expressing PpHSF5 after cultivation for one week. (B) Root length of T2 transgenic plants over-expressing PpHSF5. Three plants were measured in each biological replicate. (C) Phenotype of T2 transgenic plants from two lines over-expressing PpHSF5 after cultivation for two weeks. Seeds were transferred to soil after germination and growth on agar for five days. (D) The morphology and relative expression of T2 transgenic plants with PpHSF5 and WT after cultivation for two weeks. The number of rosettes, length and width after cultivation for two weeks. Relative expression of PpHSF5 in transgenic Arabidopsis plants carrying p35S:PpHSF5; (E) Phenotype of T2 transgenic plants over-expressing PpHSF5 after cultivation in soil for three weeks. (F) The length, width and number of rosette leaves, number of internodes and flower stalks, and the height of plants after cultivation for three weeks. (G) Phenotype of T2 transgenic plant roots over-expressing PpHSF5 after cultivation for three weeks. (H) The root length, volume and other indexes were scanned after cultivation for three weeks.

PpHSF5-OE lines exhibit enhanced thermotolerance

The thermotolerance of PpHSF5-OE lines was assayed with that of WT (Fig. 7 and Table S7). As shown in Fig. 7B and 7F, only 8.3% WT seeds germinated, whereas more than 93.3% of the transgenic seeds germinated after HS treatment 1 d. Nearly half of the WT seeds germinated after HS treatment 3 d, whereas 100% of the transgenic seeds were germinated (Fig. 7C and 7F). After HS treatment 5 d and 7 d, 68.4% and 82.6% of WT seeds germinated, respectively (Figs. 7D, 7E and 7F). Compared to WT seedlings, the PpHSF5-OE seedlings exhibited green cotyledons and vigor growth (Figs. 7C, 7D and 7E). These results suggested that the overexpression of PpHSF5 improves thermotolerance of PpHSF5-OE lines.

Figure 7 Thermotolerance of the p35S:: PpHSF5 plants.

(A) Five-day-old seedlings of wild type and the p35S:: PpHSF5 plants were treated at 46 °C for 30 min. Photographs were taken before HS treatment. (B) Photographs were taken after 1d in 22 °C. (C) Photographs were taken after 3d in 22 °C. (D) Photographs were taken after 5d in 22 °C. (E) Photographs were taken after 7d in 22 °C. (F) Comparison of germination rate among wild-type, p35S::PpHSF5 transgenetic plants after HS treatment. The number of germinated plants was counted daily after HS treatment. For three replication, more than 50 seedlings were used each lines (t-test signifificant at P < 0.05 and P < 0.01, respectively).

Discussion

Peach contains fewer HSF gene family members among several plant species

HSFs play vital roles in plant growth and defense. Through plant genome sequencing, HSF gene family members have been identified in several model organisms and more than 20 plant species (Table S8). Only a single HSF was detected in yeast, nematodes and flies (Nakai, 1999; Nover, 1996). In this study, 18 HSF genes were identified in peach, which is less than in most other plant species, but more than in tea (Camellia sinensis), strawberry (Fragaria vesca), Chinese plum (Prunus salicina) and carnation (Dianthus caryophyllus) (Hu et al., 2015; Liu et al., 2016; Qiao et al., 2015; Li et al., 2019).

HSFs in each subgroup are highly similar to each other across a variety of plants. Among these species, Class A contains the largest number of HSFs, followed by Class B, and then Class C. The same phenomenon was also observed in peach, which contained 11 HSFAs, six HSFBs and one HSFCs.

The HSF gene family expanded along with DSD in peach

The number of HSFs expanded markedly during plant evolution. The analysis of 51 representative species indicated that the HSF gene family largely expanded along with WGD during plant evolution (Wang et al., 2018). In Chinese white pear (Pyrus bretschneideri), most PbHSF expansions dated back to a recent WGD (Qiao et al., 2015). On the other hand, GmHSFs in cotton expanded along single gene duplication events (Wang et al., 2014). Here, DSD (67%) was the primary type of duplication for the HSF gene family in peach. The same phenomenon was also seen for the E3 ligase gene family in peach (Tan et al., 2019). It is probably that peach has not undergone a recent WGD (Verde et al., 2013).

HSF gene family was classified into three classes

Plant HSF proteins contain a few conserved characteristic domain (Guo et al., 2016). Generally, HSF families in plant species can be divided into three subfamilies, termed HSFA, HSFB, and HSFC (Liu et al., 2018; Wang et al., 2018). The classification of the PpHSF family was consistent with that in other plant species (tabreftabs7). Multiple sequence alignments revealed that an insertion occurred in the DBD domain near the N-terminus in the PpHSFA and PpHSFC groups. Like in other plants, the PpHSFA and PpHSFC genes contained inserted coding sequence for 21 and 7 aa in the HR-A/B region, respectively, while the HR-A/B region of PpHSFB was compact (Nover, 1996; Scharf et al., 2012). The organization, composition, number of conserved motifs in the HSFs differed among plant species (Wang et al., 2018). In Chinese whit pear, Class A in PbHSFs contained the most conversed motifs, followed by class B and then class C (Qiao et al., 2015). In this study, the number of motifs in the different classes was consistent with those in Chinese white pear. This also showed that members of the same class often have similar sequence structures in peach. For example, motif 5 was present only in PpHSFA and PpHSFC, while all Class B and Class A HSFs contain motif 4. The presence of these motifs may lead to functional group specificity. The similar classifications of HSF families in diverse plants showed that the HSF family was highly conserved during long-term evolution.

PpHSF5 acts as repressor of organ size in plants

In plants, organ size is primarily controlled by internal developmental signals (Mizukami, 2001; Dubrovsky et al., 2006; Spradling, Drummond-Barbosa & Kai, 2001; West, Inzé & Beemster, 2004). Previous research in the model organism Arabidopsis thaliana indicates that plant hormones and transcription factors, including HSFs, play crucial roles in growth and development (Petricka, Winter & Benfey, 2012; Begum, Reuter & Schoffl, 2013). HSFs as key transcription factors protect plants from various abiotic stresses and then participate in the growth and development (Guo et al., 2016). For example, OsHsfA1a, OsHsfA1b and OsHsfA1d are the main positive regulators of gene expression on heat stress-responsive, and four HSFA proteins play significant roles in gene expression of plant growth and development (Yoshida et al., 2011). In poplar (Populus trichocarpa), the transcripts of three PtHsfs in the B4 subfamily (-B4b, -B4c and -B4d) were maintained at higher levels during the leaf expansion stages (Liu et al., 2019). In carnation (Dianthus caryophyllus), five DcaHsfs, namely DcaHsf-A1, A2a, A9a, B2a, B3a, were involved in early flowering stages (Li et al., 2019). Transgenic Arabidopsis plants overexpressing AtHSFB4 contained massively enhanced levels of AtHSFB4 mRNAs and exhibited shorter roots (Begum, Reuter & Schoffl, 2013). In this study, overexpression of 35S:PpHSF5 in Arabidopsis resulted in not only shorter roots but also in lesser root volume and fewer lateral roots and root forks compared to WT.

The root system of a plant is instrumental to its growth and productivity because it is responsible for the extraction of water and mineral nutrients from the soil and their transport to aboveground parts of the plant (Hochholdinger & Feix, 1998). In this study, the 35S:PpHSF5 transgenic lines produced smaller aerial organs compared with WT. For example, the size (length and width) of rosette leaves were smaller than WT two and three weeks after transplanting, while the number of rosette leaves was not affected. The height of the overexpression lines was significantly lower than that in WT, while the number of internodes was not. Overexpression of OsHsfA2e in rice caused a dwarf phenotype (Yokotani et al., 2007). In plants overexpressing BhHsf1, the reduced organ size was mainly attributed to decreased cell proliferation (Zhu et al., 2009). The overexpression of PpHSF5 in peach suggested that the dwarf phenotype of transgenic plants was caused by shorter internodes.

It is still unknown how PpHSF5 regulates root and aerial organs development. PpHSF5 is homologous to AtHSFB4 and thus may play similar roles in root development. Confocal laser scanning of roots in AtHSFB4-overexpression transgenic lines showed that ectopic division of the lateral root cap cells (LRC) occurred (Begum, Reuter & Schoffl, 2013). Previous studies indicated that auxin acts in the production of lateral root primordium (LR) (Casimiro et al., 2003; West, Inzé & Beemster, 2004). In the promoter of PpHSF5, there are three cis-acting regulatory elements that contain the auxin-inducible TGACG-motif. Two auxin-inducible TGA-box elements in the GmGH3 promoter were strong binding sites of plant nuclear proteins and improved the auxin inducibility of the GmGH3 promoter (Zhan-Bin Liu et al., 1994). Moreover, the HS assays indicated PpHSF5-OE lines exhibited enhanced thermotolerance compared to WT. Similarity results were observed in transgenic Arabidopsis plants with AtHsfA3 and rice plants with OsHsfA2e (Ogawa, Yamaguchi & Nishiuchi, 2007; Yokotani et al., 2007). Therefore, PpHSF5 might be as a responsive factor for temperature change and involved in auxin signal transduction due to the TGA motifs in its promoter and might serve to negatively regulate root elongation and lateral root development, ultimately affecting the growth of aboveground parts of the plant.

Conclusions

In this report, 18 PpHSF genes were discovered in peach and found to be nonuniformly distributed on the peach chromosomes. The PpHSF family could be classified into three classes (PpHSFA, PpHSFB and PpHSFC) through multiple alignment, motif analysis and phylogenetic comparison. The expansion of the HSF gene family in peach occurred through DSD (67%) and WGD (33%). PpHSF5 was expressed in diverse tissues and organs of the peach cultivar ‘Zhongyoutao 14’, with higher levels in young vegetative organs (leaf and apex). Transgenic Arabidopsis lines overexpressing PpHSF5 showed massively enhanced levels of PpHSF5. Ectopic expression PpHSF5 repressed the length and number of roots, length and width of rosette leaves, and the height of plants, and enhanced thermtolerance in Arabidopsis after heat stress treatment. Our results further supplied functional and annotation information of the HSF gene family in general and revealed potential roles, outside of the response to heat stress, for PpHSF5 during plant development.

Supplemental Information

Supplemental Information 1 The value of FPKM of PpHSF genes between four periods in ‘Zhong Youtao 14’ and list of primers used in this study

Click here for additional data file.

Supplemental Information 2 The characteristics of HSF genes in peach

Click here for additional data file.

Supplemental Information 3 Prevelance of duplication modes in thePpHSF family

DSD: dispersed gene duplication; WGD: whole-genome duplication.

Click here for additional data file.

Supplemental Information 4 Motif sequences identified by MEME tools in PpHSFs

Click here for additional data file.

Supplemental Information 5 The data of quantitative real-time PCR

Click here for additional data file.

Supplemental Information 6 Phenotypic data of over-expressing PpHSF5 and wild Arabidopsis plants

Click here for additional data file.

Supplemental Information 7 The germination rate of WT and OE Arabidopsis thaliana after HS treatment

Click here for additional data file.

Supplemental Information 8 The number of HSF gene family members in different plant species

Click here for additional data file.

Supplemental Information 9 Gene structures of the 18 HSF gene family members in peach

Predicted Coding Sequences (CDS) are in yellow, introns are a flat line, and the untranslated regions (UTRs) are shown in blue.

Click here for additional data file.

The authors would like to thank Anita K. Snyder for critical reading the manuscript.

Additional Information and Declarations

Competing Interests

Author Contributions

Data Availability

The authors declare there are no competing interests.

Bin Tan conceived and designed the experiments, performed the experiments, analyzed the data, prepared figures and/or tables, authored or reviewed drafts of the paper, and approved the final draft.

Liu Yan and Huannan Li performed the experiments, analyzed the data, prepared figures and/or tables, authored or reviewed drafts of the paper, and approved the final draft.

Xiaodong Lian, Jun Cheng and Wei Wang analyzed the data, prepared figures and/or tables, authored or reviewed drafts of the paper, and approved the final draft.

Xianbo Zheng and Xiaobei Wang performed the experiments, prepared figures and/or tables, authored or reviewed drafts of the paper, and approved the final draft.

Jidong Li, Xia Ye, Langlang Zhang and Zhiqian Li performed the experiments, authored or reviewed drafts of the paper, and approved the final draft.

Jiancan Feng conceived and designed the experiments, analyzed the data, prepared figures and/or tables, authored or reviewed drafts of the paper, and approved the final draft.

The following information was supplied regarding data availability:

Raw data is available in the Supplemental Files.

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
