# Peer review of "Genome-wide identification of HSF family in peach and functional analysis of PpHSF5 involvement in root and aerial organ development"

_PeerJ, doi:10.7717/peerj.10961_

## Round 0.1 · original submission · Major Revisions

The reviewers have requested substantial changes to your manuscript. We will allow you to resubmit it if you substantially improve the manuscript, as explained in the reviewer's comments.

Reviewer 1 ·

Basic reporting

no

Experimental design

no

Validity of the findings

no

Additional comments

In the manuscript, 18 Hsfs were indentified from peach through the genomic DNA sequences, and their biophysical properties and expression profiles in some tissues and preliminary functional of PpHsf5 in root development were analyzed. The results are some informative, but overall, the paper seems to be so easy, the experiments of functional analysis were not enough to verify the results. So, the following problems must be solved.
1. According to the results of functional analysis in manuscript, author thought that PpHSF5 may depress root development, so, in the experiment of tissue expression of Hsfs, 18 PpHsfs or at least 6 PpHsfBs should be added, especially in roots, so as to show the specificity of PpHsf5.
2. Most Hsf are responsive to heat stress, so in the manuscript, 18 Hsf member expression level after heat stress should be added, so as to show the basic expressing characteristics of PpHsf family.
3. The data of qPCR in Figure 5 seems to be calculated while the expression of Apical was set as 1, if so, the standard error in figure was marked incorrectly, for example the standard error in “Apical” should be 0, all these should be introduced clearly in Material.
4. In figure7, not only root but also rossete of two transgenic Arabidopsis lines were weak than WT, so it`s necessary to add the phenotypes of more plants together in plate of twoweeks and eight weeks, three plants are not enough to verify the results. How many transgenic lines were obtained? Are all lines single-site insertion or not?
5. In figure7-C, no PpHSF5 expression can be detected in WT because WT Arabidopsis has no PpHSF5 gene, How did author calculate and get the gene PpHSF5 expression level in two transgenic lines? Which was set as 1?
6. The qRT-PCR of PpHsf5 expression and length、numbers and activities of Arabidopsis roots both two weeks and eight weeks should all be analyzed, add the experiment of some relative Hsp expression in roots (example Hsp90 et al) of transgenic Arabidopsis lines and WT. There is no experimental data of Semi-RT-PCR to analyze positive transgenic lines of PpHsf5.
7. Some error should be corrected such as figure 6 : F: Overlay of F and E.
8. Add the references about plant Hsf of 2019 and 2020.

Reviewer 2 ·

Basic reporting

1. Tan et al. identified HSF genes from the peach genome and conducted functional assays of PpHSF5 to determine its role in development.

2. The analyses are sound, yet a clearer explanation of the rationale behind certain analysis is needed. For example, Tan et al. didn't explain why they chose to focus on PpHSF5, which is actually an important part of the logic flow.
Besides, I'm not sure I get the point of the intron analysis. I suggest moving Fig 2 to the supplementary materials and delete the first sentence from the abstract as well.

3. The language can be improved presumably with the help of a native English speaker. Present tense is preferred for delivering results, but of course people do both.

Experimental design

4. It is necessary to define Class A/B/C before using the terms. Alternatively, moving Fig 3B before Fig 3A will be helpful, which is how I get to understand Class A/B/C.
I also suggest making the current Fig 4 the third panel of Fig 3.

Validity of the findings

5. What was PpHSF5 compared to to obtain the relative expression level in the RT-PCR assay?

6. More detailed captions for Fig 3 and Fig 5 are needed.

Additional comments

It would be great if the discussion part could be shortened. Currently it is quite lengthy (4.5 pages) compared to results (4 pages), yet more of speculations than real solid findings. It would be extremely helpful to add more text to elaborate the logic flow in the results section.

---

## Round 0.2 · Major Revisions

The reviewers addressed several concerns. Please make substantial improvement to your manuscript to reach the standard of PeerJ.

Reviewer 1 ·

Basic reporting

no

Experimental design

no

Validity of the findings

no

Additional comments

1. In this study, the authors mainly focused on the effect of PpHSF5 gene on plant growth and development, especially on root growth and development, they gave expression of different organs from ‘Zhongyoutao 14’ and added the transcript profile of 18 PpHSFs during shoot development of ‘Zhongyoutao 14’, but in these results, no any expression of roots, so it`s less relative to later experiments and title. Please add the roots expression of different development stages of ‘Zhongyoutao 14’.
2. Add the more plants roots phenotypes of two transgenic lines of two weeks
together in one plate and eight weeks, three plants each line are not enough to verify the results, and there is no evidence to prove these plants come from the same plate.

Annotated reviews are not available for download in order to protect the identity of reviewers who chose to remain anonymous.

·

Basic reporting

The author analyzed the HSF gene family of pear genome, and studying the functional of hsf5 in regulating root length development.

Experimental design

Authors analyzed 18 members of HSF gene family, promoter elements and further analyzed the negative regulatory effect of HSF5 on root development

Validity of the findings

Authors analyzed 18 members of HSF gene family, promoter elements and further analyzed the negative regulatory effect of HSF5 on root development

Additional comments

1, Compared with other species, which members of HSF gene family in Pear genome had evolved under the pressure of artificial selection?
2, Previous studies have found that HSF gene family are mainly involved in the regulation of gene expression under stress conditions. In the paper, author only focuses on the temperature sensitivity pear shoot growth, but does not involve other stress response. However, author selected hsf5 gene for functional identification is also aimed at the regulation of root length. Why? The stress treatment is too little, so little informations can be got from the process of stress response to HSF gene family.

Reviewer 4 ·

Basic reporting

The level of the English language used throughout the manuscript is of a low standard, to being very poor. Many sentences are so poorly worded that they do not make any sense to the reader.

Background to the area is average at best. A lot background is required for ease of understanding for the reader.

Figures are also of an average standard. Tables add little.

Not a self-contained article as not enough background information is provided, or is too poorly explained to adequately understand.

Experimental design

Very standard largely bioinformatic based study with little supporting experimental work provided.

Research question is defined, but not overly relevant or meaningful.

Investigative rigour is lacking in this article. 18 gene family members identified, yet only one experimentally validated - not acceptable rigour and why this family was selected for further analysis over the other family members is also lacking. Only two Arabidopsis transformant lines generated and analysed - not acceptable scientific rigour.

Bare minimum methodological detail provided - one would struggle to reproduce many aspects of this study due to lacking detail.

Validity of the findings

Not an appropriate level of experimental validation performed to support the provided bioinformatic data.

RT-qPCR assessment of one gene family member from 18 identified is not acceptable

Required number of transformant lines are also lacking for the Arabidopsis work.

Discussion and Conclusion are far too general in their nature and do not tailor accordingly to the specific of the study.

Some claims made by the authors are not supported by the experimental evidence and/or rigour of the reported results.

Additional comments

See above comments addressing criteria 1 to 3.

---

## Round 0.3 · Major Revisions

Please make a substantial revision to your manuscript based on the reviewer's comments.

Reviewer 1 ·

Basic reporting

no

Experimental design

no

Validity of the findings

no

Additional comments

1. Most Arabidopsis plant of Hsf transgenic showed dwarf and develop slowly, but thermotolerances were enhenced (Ogawa D et al. J Exp Bot, 2007, 58: 3373–3383). Meanwhile,From the Heatmap of transcript levels of HSF genes in peach in this paper (Fig. 3), with the growth stages of temperature-sensitive peach shoot development, PpHSF5 were remarkably upregulated, which means PpHSF5 is strong heat-response gene, so the thermotolerance function of gene must be performed firstly.
2. In Fig.6, the legend was not coherent with the photos. To ensure the consistence, the seedlings of different transgenic lines should be planted in one culture pot.To validate the experimental results, more seedlings than one should also be cultured in every lines. The data of Figure 6B/D/F/H is no sense, the better phenotypic photos can exhibit the results.

Reviewer 2 ·

Basic reporting

Tan et al. elucidated the rationales behind analyses at multiple places, which makes the logic flow clear and coherent.

Most of my concerns have been handled except for a few minor points. There is no response to my comments in the rebuttal doc. I suggest accepting this manuscript.

Experimental design

The authors rearranged the figures as I suggested, which are currently easy to follow.

Validity of the findings

The study is more complete with the newly added functional experiments.

The authors may want to add the name of statistical tests and their power for the relative expression analysis. For now, I can't tell if some conclusions are qualitative or quantitative, and even if some of the claims hold true statistically.

Additional comments

There are missing caption for newly added figures, e.g. Fig 6E-H.

---

## Round 0.4 · Major Revisions

Hello. I have taken over as Academic Editor for your manuscript. Thank you for addressing the previous reviewer concerns. I have read through the manuscript and have found some additional items that need to be addressed:

1) The peach genome already has functional annotation. Indeed. searching through the annotation for heat shock proteins reveals over 100 genes. Did your search reveal any heat shock genes not already annotated as such? Why did you only find 18 when there are so many already annotated (different families or something wrong with the search)? Explain the value added by your search beyond what was already available. See ftp://ftp.bioinfo.wsu.edu/species/Prunus_persica/Prunus_persica-genome.v2.0.a1/functional/ (I assume this is on phytozome also but it is down for maintenance today)

2) Neighbor joining is an inferior method for phylogenetic reconstruction. Please use a maximum likelihood based method. You also need to show bootstrap support for your tree.

3) Your transgenic results show that PpHSF5 represses growth and development. Thus, the interpretation from the heatmap and qRTPCR that elongation might be caused by increased expression of PpHSF5 (line 279) and that it might play a vital role in the sprouting process (line 285) do not make sense to me.

Minor:

4) list the version of the peach genome that you were using (not just phytozome version)

5) The sentence (line 18) "While most..." is a sentence fragment, not a sentence.

Reviewer 1 ·

Basic reporting

no

Experimental design

no

Validity of the findings

no

Additional comments

My concerns have been handled.

---

## Round 0.5 · Minor Revisions

Hi, thanks for your response and modifications. There are still a couple of the issues that need attention.

Regarding my point 1 (18 genes vs over 100) thank you for clarifying heat shock protein vs heat shock factor. There are 20 genes annotated as heat shock factor in peach genome annotation, v2.0. Are all of your 18 among these 20 or did you find some that are not currently annotated? And for the at least 2 that are currently annotated as heat shock factors but that are not in your list, are those not true heat shock factors or did your search miss them? As I requested before, an explanation of how your search improves on current annotation is needed. A table comparing your annotation to the 20 in the current annotation would be helpful.

Regarding my point 4 (peach genome version): I still don't see that the version of the peach genome that you used is stated. I appreciate you giving the URL but also give the version of the peach genome, because the version hosted at phytozome (and at that URL) will change over time. Did you use 1.0, 2.0, or 2.1? Also I note on the phytozome page that they request that you cite the sequencing paper. You have that cited in the discussion but it should also be cited in the methods and maybe main text when you first bring up the peach genome sequence.

The International Peach Genome Initiative., Verde, I., Abbott, A. et al. The high-quality draft genome of peach (Prunus persica) identifies unique patterns of genetic diversity, domestication and genome evolution. Nat Genet 45, 487–494 (2013). https://doi.org/10.1038/ng.2586

---

## Round 0.6 · accepted · Accept

Thank you for the updates to your manuscript and for submitting your work to PeerJ